# The Impact of Statin Use on Sepsis Mortality: A Systematic Review and Meta-Analysis

**DOI:** 10.3390/medicina61091563

**Published:** 2025-08-30

**Authors:** Constantinos Philippou, Constantinos Tsioutis, Maria Tsiappari, Nikolaos Spernovasilis, Dimitrios Papadopoulos, Aris P. Agouridis

**Affiliations:** 1School of Medicine, European University Cyprus, Nicosia 2404, Cyprus; cp211877@students.euc.ac.cy (C.P.); mt192099@students.euc.ac.cy (M.T.); nikspe@hotmail.com (N.S.); d.papadopoulos@euc.ac.cy (D.P.); a.angouridis@euc.ac.cy (A.P.A.); 2Department of Infectious Diseases, German Medical Institute, Limassol 4108, Cyprus; 3Department of Internal Medicine, German Medical Institute, Limassol 4108, Cyprus

**Keywords:** statins, HMG-CoA reductase inhibitors, sepsis, bacteremia, mortality

## Abstract

*Background and Objectives*: Statins are among the most prescribed medications globally, primarily due to their potent lipid-lowering capabilities. This systematic review aims to identify, synthesize and evaluate current evidence regarding the potential protective effects of statins on sepsis mortality. *Materials and Methods*: A thorough and comprehensive database search was conducted in PubMed and Cochrane Library until 30 January 2025. Randomized control trials (RCTs) and cohort studies evaluating the effect of statin use on sepsis mortality were included. Risk-ratios (RRs) and 95% confidence intervals (CIs) were calculated. Statistical analysis and forest plot generation were performed using RevMan 5.4. Risk of bias was assessed using the RoB-2 and NOS tools. *Results*: A total of 49 studies were identified following application of the PRISMA guidelines. Of these, 16 studies were RCTs and 33 were cohort studies. The pooled analysis of RCTs demonstrated a non-significant 10% reduction in mortality in statin users (RR: 0.90, 95% CI 0.80–1.01). The pooled analysis of cohort studies showed that statin users have a 21% significantly reduced mortality risk (RR: 0.79, 95% CI 0.72–0.86). For the de novo statin users vs non-statin users, pooled analysis demonstrated a significant 25% reduction in mortality (RR: 0.75, 95% CI 0.69–0.81). The pooled analysis for the continuation of prior statin use vs discontinuation of statin use indicated 52% lower mortality in statin users who continued the use of statins (RR: 0.48, 95% CI 0.25–0.92). The pooled analysis of prior statin use and continuation of statins vs non-statin use revealed a significant 23% lower risk in statin users compared with non-statin users (RR: 0.77, 95% CI 0.69–086). *Conclusions*: According to our findings, statin use among septic patients is associated with a reduction in mortality, suggesting that statins may offer a beneficial therapeutic effect in the clinical setting. Clinicians may consider the continuation or potential incorporation of statin use as an additional regimen in the treatment of septic patients.

## 1. Introduction

Sepsis is a severe and acute life-threatening organ dysfunction resulting from an uncontrolled immune response of the host to infection. High morbidity and mortality are closely associated with sepsis and septic shock, a subset of sepsis with circulatory, cellular or metabolic anomalies [1]. Early recognition and intervention are of paramount importance in order to reduce mortality since the pathophysiology behind sepsis is complex, and its signs and symptoms are nonspecific. Early fluid resuscitation, broad spectrum antibiotics, vasopressors, oxygen supple-mentation and corticosteroids in certain cases are among the standard regimens of sepsis [2]. While these interventions are essential and beneficial, sepsis mortality rates still remain high [3]. As a result, it is crucial to explore alternative agents, such as statins, and examine whether those agents exert beneficial effects on sepsis or not.

Statins represent the most prescribed drugs worldwide, since they are the lipid-lowering drugs of choice [4]. Also known as 3-hydroxy-3-methylglutaryl-Coenzyme A (HMG-CoA) reductase inhibitors, statins act on the synthesis of cholesterol in hepatocytes by competitively inhibiting the HMG-CoA reductase enzyme [5]. Since statins improve the lipid profile by reducing low-density lipoprotein (LDL-C) levels, they are the key contributors in the prevention and management of Atherosclerotic Cardiovascular Disease (ASCVD). This is of utmost importance because ASCVD continues to be the primary cause of mortality globally [6,7].

Although the primary goal of statins is to reduce LDL-C levels, research has indicated that statins have also other properties called “pleiotropic effects” [4]. These include anti-inflammatory and anti-thrombotic properties, the suppression of smooth muscle cell proliferation and apoptosis and the inhibition of macrophage migration and activation [8].

While significant research has been conducted on statins and sepsis independently, a crucial gap remains in understanding the impact of statin use on sepsis mortality. Since current evidence gives mixed results and conclusions regarding the efficacy of statins in critically ill patients [9,10,11,12,13,14], it would be interesting to perform an updated systematic review and meta-analysis on this topic. The aim of this systematic review is to investigate whether the outcomes of patients who suffer from sepsis can be benefited from statin use.

## 2. Materials and Methods

### 2.1. Study Design

We performed qualitative and quantitative syntheses of RCTs and cohort studies to determine and evaluate the role of statins on sepsis mortality. This systematic review has been registered in the International Prospective Register of Systematic Reviews (PROSPERO), with a Prospero ID (CRD420251026667), and adhered to the Preferred Reporting Items for Systematic Reviews and Meta-Analyses (PRISMA) 2020 statement (PRISMA Statement, Ottawa, ON, Canada) [15].

### 2.2. Search Strategy

An extensive systematic literature search was conducted through PubMed and Cochrane library databases until 30 of January 2025, using combinations of the following keywords: “statin”, “atorvastatin”, “fluvastatin”, “lovastatin”, “pitavastatin”, “pravastatin”, “rosuvastatin”, “simvastatin”, “HMG-CoA reductase inhibitors”, “sepsis”, “mortality” and “bacteraemia”.

### 2.3. Eligibility Criteria

Eligibility criteria followed the PICOS (population, intervention, comparators/controls, outcomes, and study design) study question format, as described in Table 1.

Randomized control trials and case-control and cohort studies using an adult population and conducted in English language have been considered for inclusion in the systematic synthesis. Studies were not considered eligible if they met one of the following exclusion criteria:Articles not in the English language;Animal studies;Non-original articles (review, medical hypothesis, letter to the editor, etc.);Systematic reviews and meta-analyses;Duplicated papers;Studies that could not be obtained;Adolescents and children with sepsis;Absence of established data of sepsis;Absence of established data of statin use;

In addition, we have also hand-searched the reference lists of any relevant reviews/articles for further relevant material.

### 2.4. Data Extraction

During data extraction, the title and abstract of each record were scanned for eligibility. Following the initial database search, all search results were manually screened for duplicates. The full text was then assessed to determine whether the selected studies met the inclusion criteria. The following data were extracted: first author, publication year, country where the study was conducted, study type, characteristics of the participants of each study, information regarding the characteristics of each statin used and the relevant information concerning mortality.

### 2.5. Assessment of Bias

For the assessment of the eligible RCTs, the Revised Cochrane risk-of-bias tool for randomized trials (RoB-2) was used. In addition, the measure of blinding was assessed using the ROB2 tool. The Newcastle–Ottawa Scale (NOS) was used and applied individually for the assessment of the remaining cohort studies.

### 2.6. Quantitative Analysis

Following a thorough examination of the included articles, meta-analyses of the included RCTs and cohort studies were conducted. Risk-ratios (RRs) and 95% confidence intervals (CIs) were also calculated. The statistical analyses were conducted using Review Manager (RevMan) version 5.4 (The Cochrane Collaboration, Copenhagen, Denmark, 2020), and forest plots were generated accordingly. Heterogeneity was assessed using the I^2^ test, with values of 25%, 50% and 75% interpreted as low, moderate, and high heterogeneity, respectively. The random-effects model was used when heterogeneity was high, and the fixed-effects model was used in the performed meta-analyses when the heterogeneity was low. A *p*-value of <0.05 was considered statistically significant.

### 2.7. Assessment of Publication Bias

Publications bias was assessed through visual inspection of funnel plots, which plotted effect estimates against their standard error on a reversed scale. These plots were generated using Review Manager (RevMan) version 5.4 (The Cochrane Collaboration, Copenhagen, Denmark, 2020). Asymmetry in the funnel plots was interpreted as potential evidence of publication bias, while symmetry of the plot suggested low publication bias.

## 3. Results

### 3.1. Study Selection

The search and selection process are presented in Figure 1, in the PRISMA 2020 flowchart. The initial database search identified a total of 727 articles from two different databases, including 583 articles from PubMed and 144 from Cochrane Library. After the deduplication process, 674 articles were left to be screened. Titles and abstracts were then screened for relevance, and the remaining 81 articles were screened for eligibility. During the full text article screening, 7 additional articles were identified, raising the number of the assessed articles for eligibility to 88. After the exclusion of ineligible articles, 49 studies were judged as eligible for the systematic review.

### 3.2. Study and Population Characteristics

The characteristics of the included studies are shown in Table 2 and Table 3. Among the 49 eligible studies, 16 were RCTs [16,17,18,19,20,21,22,23,24,25,26,27,28,29,30,31] involving 2876 participants and 33 were cohort studies [32,33,34,35,36,37,38,39,40,41,42,43,44,45,46,47,48,49,50,51,52,53,54,55,56,57,58,59,60,61,62,63,64] involving 267,368 participants, respectively. The studies were published between 2006 and 2024 and enrolled participants of both genders. In total, the current review included 270,244 participants, of whom 90,515 received statin therapy, while the remaining 179,729 did not. The study populations in the eligible studies were drawn from different geographical regions, including Africa, Asia, Europe, North and South America and Oceania. All studies were conducted in clinical settings, primarily hospitals and intensive care units (ICUs). The majority of the included RCTs (15 out of 16) compared the effect of the de novo use of statins against the effect of a placebo. One RCT compared the outcomes of continuing prior statin use against those of discontinuing prior statin use. In the included RCTs, six administered Simvastatin, three utilized Rosuvastatin, six provided Atorvastatin and one used Pravastatin. All the RCTs provided information regarding the dose of each statin used. Information was also provided regarding the underlying conditions that led to sepsis. Specifically, two of the RCTs focused on bacteremia, six on various types of pneumonia, two on septic shock and the remaining studies addressed sepsis without further specification. Regarding the cohort studies, 29 of 33 compared the effects of continuing the prior use of statins against non-statin users, 2 compared the de novo statin use against outcomes of non-statin users, 1 compared the effects of continuing the prior statin use against discontinuing and 1 did not specify if there was any prior use of statins in the comparison group. None of the cohort studies specified the type or dosage of statins. Underlying and/or associated conditions in the cohort studies were as follows: bacteremia as the reason that led to sepsis (*n* = 9), candidemia (*n* = 1), multiple organ dysfunction syndrome (*n* = 1), acute respiratory distress syndrome (ARDS, *n* = 1), sepsis-induced coagulopathy (*n* = 1), chronic kidney disease (*n* = 1), severe sepsis and septic shock (*n* = 5), whereas 14 studies addressed sepsis without further specification. Of note, the majority of the included RCTs reported 28-day or 30-day mortality (10 studies), 1 study reported 14-day mortality, 1 study reported 60-day mortality and 1 study reported 90-day mortality, while 3 studies did not mention how mortality was evaluated. In the 33 included cohorts, 13 studies reported 28-day or 30-day day mortality, while 20 studies did not mention details regarding mortality.

### 3.3. Analysis of Evidence

Pooled analysis of the included RCTs is shown in Figure 2. Statin users have a 10% lower mortality risk in comparison with non-statin users, but this was not significant (RR: 0.90, 95% CI 0.80–1.01, *p* = 0.07, I^2^: 3%). Regarding the pooled analysis of the cohort studies (Figure 3), statin users have a significant 21% lower mortality risk in comparison with non-statin users (RR: 0.79, 95% CI 0.72–0.86, *p* < 0.00001, I^2^: 93%).

As far as publication bias is concerned, a relative asymmetry is noted in the funnel plot primarily due to the study by Viasus et al. [27], while similarly, the relative asymmetry among cohort studies was mainly attributable to the study by Schurr et al. [52] (Figure 4 and Figure 5). Nevertheless, the funnel plots for the comparison of statin vs. non-statin users on mortality risk revealed low overall publication bias.

### 3.4. Subgroup Analyses

We performed four additional meta-analyses, three on cohort studies and one for RCTs on the “Exposure and Comparator”. In Figure 6, the pooled analysis from three cohort studies, including 2778 de novo statin users and 24,181 non-statin users, demonstrated a significant 25% lower risk of mortality in statin users (RR: 0.75, 95% CI 0.69–0.81, *p* < 0.00001, I^2^ = 0%).

In Figure 7, the pooled analysis from three cohorts, including 435 patients with prior and continuation of statin therapy and 777 patients with prior statin use and the discontinuation of the statin therapy, showed a significant 52% lower mortality risk in continuing statin users compared with discontinuing statin users (RR: 0.48, 95% CI 0.25–0.92, *p* = 0.03, I^2^ = 66%).

The pooled analysis from 28 cohorts with prior statin use and the continuation of statins (85,914 septic patients) and non-statin users (183,477 septic patients) showed a significant 23% lower risk in statin users compared with non-statin users (RR: 0.77, 95% CI 0.69–086, *p* < 0.00001, I^2^ = 93%), (Figure 8).

Finally, the pooled analysis from 15 RCTs showed a non-significant 10% lower risk in de novo statin users compared to the placebo (RR: 0.90, 95% CI 0.80–1.02, *p* = 0.10, I^2^ = 2%), (Figure 9).

### 3.5. Sensitivity Analyses

Due to high heterogeneity, we also performed sensitivity analyses for Figure 3 and Figure 8. After excluding the studies identified as outliers in Figure 2 and Figure 8, our analysis reached 0% heterogeneity for both syntheses, by use of the fixed effects model (Appendix A). In brief, regarding Figure 3, similar reductions in mortality risk were observed (RR: 0.73, 95% CI 0.66–080, *p* < 0.001, I2 = 0%) (Appendix A), while regarding Figure 8, a 14% mortality risk reduction was observed (RR: 0.86, 95% CI 0.84–089, *p* < 0.001, I2 = 0%) (Appendix A).

### 3.6. Quality Appraisal

All of the included studies were assessed for their quality. The RCTs were appraised using the RoB-2 tool, and the cohort studies were assessed using the NOS tool. In Figure 10 and Figure 11, the detailed methodological quality of individual RCTs is shown. Most studies were rated as low risk of bias across all domains, though some of them showed some concerns or high risk. Overall, they demonstrated acceptable methodological quality, supporting the reliability of their findings. The quality assessment of the cohort studies is shown in Table 4. Most of the cohort studies (32 from 33) were considered high quality, with only one study rated as low quality.

## 4. Discussion

The present systematic review and meta-analysis evaluates the impact of statin therapy on survival outcomes in septic patients. According to our findings, statin therapy exerts a protective effect on septic patients by reducing the overall risk of mortality. More specifically, the pooled synthesis of the included cohort studies illustrated a significant reduction in mortality among statin users, while the pooled analysis of RCTs demonstrated a non-significant alteration of the mortality risk. Similar results were seen during the sensitivity analysis process.

Over the past decade, attention has been directed towards the pleiotropic effects of statins. Previous meta-analyses have explored the association between statin therapy and clinical outcomes in septic patients, in an effort to provide valuable evidence to determine whether statins offer a beneficial effect in this clinical setting. A meta-analysis of seven RCTs, conducted in 2015 [65], concluded that statin therapy does not improve the mortality outcome of septic patients. In the same year, a meta-analysis of four RCTs [9] showed no difference in terms of mortality rates between statin and placebo users. On the contrary, Janda et al. [10], in 2010, reported a protective effect of statins in patients with sepsis, while Falagas et al. [11] reached the same conclusions in 2008. This discrepancy in results continued with the mixed conclusions provided by several other meta-analyses [12,13,66].

The most recent meta-analysis, published in 2019 by Pertzov et al. [14], further evaluated the role of statins in the treatment of sepsis. The pooled analysis of 14 RCTs revealed that statin therapy did not significantly reduce the 30-day all-cause mortality compared to the placebo (RR: 0.96, 95% CI 0.83– 1.10, *p* = 0.56) [14]. Additionally, no benefit was seen in preventing progression from sepsis to severe sepsis or septic shock (RR: 0.53, 95% CI 0.19–1.48). In the subgroup analysis of patients with severe sepsis, there was again no benefit observed in terms of hospital mortality (RR: 0.97, 95% CI 0.84–1.12). Concerning the secondary outcomes, statin therapy was found to be associated with a reduction in the need for mechanical ventilation. Nevertheless, the certainty of the evidence supporting this finding was very low, mainly due to the small number of events.

### 4.1. Pathophysiological Mechanisms of Statin Action on Sepsis

In an attempt to clarify the role of statins in sepsis mortality, understanding the underlying pathophysiological mechanisms by which statins influence the progression of disease is equally important. Concerning the pleiotropic effects of statins (Figure 12), several studies have identified a reduction of the inflammatory process, neuroprotective properties and improvements in kidney function. Inhibition of the enzyme HMG-CoA reductase reduces the mevalonate pathway intermediates like farnesyl, which impairs the prenylation of GTPases and lowers the pro-inflammatory signaling [67]. Additionally, statins increase the stabilization of atherosclerotic plaques, exert beneficial actions on endothelial functionality, offer antioxidant, anti-inflammatory and immunomodulatory effects and provide profitable antiplatelet properties [4]. It has been found that the improvement in endothelial vasoreactivity is mainly linked to reduced oxidative stress and drug-induced vasorelaxation. Statins help restore the balance between nitric oxide (NO) and reactive oxygen species (ROS). They enhance endothelial NO synthase (eNOS) activity and NO bioavailability, thus improving vasodilation and endothelial function while reducing the oxidative degradation of NO [68]. Moreover, experimental animal studies have determined that statin therapy not only attenuates the production of superoxide anion (O−2) and NADPH oxidase but also decreases the endogenous peroxides and peroxidase activity. In combination with the production of heme oxygenase, an enzyme with antioxidant properties, it has been established that statins may have antioxidant properties [68,69]. Other than that, statin use has been shown to minimize the levels of inflammatory markers. These include, among others, high-sensitivity C-reactive protein (hs-CRP), fibrinogens, serum amyloid A, von Willebrand factor (vWF) and platelet-activating factor acetylhydrolase (PAFAH). This is accompanied by the attenuation of several proinflammatory cytokines, such as tumor necrosis factor-α (TNF-α), interferon-γ (IFN-γ) and Interleukin 6 and 8 (IL-6, IL-8), by suppressing NF-κB and TLR4 activation, providing clinical evidence of the anti-inflammatory and immunomodulatory effects of statins [8,70]. Statin treatment may also have antiplatelet properties by inhibiting the aggregation of leukocytes. This is achieved by decreasing tissue factor expression and increasing thrombomodulin activity, helping prevent microvascular thrombosis and ischemia [71]. Additionally, its neuroprotective effects, mediated through increased eNOS expression and Akt activation and phosphorylation, along with its renoprotective effects, such as the reduction of proteinuria, provide a broader non-cardiovascular benefit of statin therapy [4,72,73].

Statins are orally administered and undergo substantial first-pass hepatic uptake via OATP1B1, with clearance mainly through biliary excretion with variable renal contribution depending on the agent [74]. Metabolism is agent-specific; simvastatin, lovastatin and atorvastatin are CYP3A4 substrates, fluvastatin is mainly CYP2C9, while pitavastatin, rosuvastatin and pravastatin have minimal CYP metabolism. The half-life of statins is again class-specific, making evening dosing for short-acting and flexible timing for longer-acting ones preferable. All statins are generally very well tolerated [4]. Myalgia, headache and gastrointestinal discomfort are the most common complaints, while myopathy, rhabdomyolysis and transaminase elevation represent significant adverse effects. Regarding drug–drug interactions, CYP3A3 inhibitors (e.g., macrolides, protease inhibitors) and grapefruit juice increase exposure and toxicity risk [74].

Overall, there are no major concerns regarding potential interactions between statins and standard sepsis treatment. The only exceptions concern macrolides and azoles, which can raise statin levels causing toxicity. In patients with acute liver injury or rhabdomyolysis risk, statin use should be deferred until renal and hepatic function is within normal function [75].

### 4.2. Strengths

Before drawing final conclusions, it is important to consider both the strengths that support the outcomes of this review and the limitations that may influence the interpretation of our findings. The comprehensive and structured search strategy, which we have conducted across multiple databases, ensured the inclusion of all relevant studies, thus minimizing selection bias and increasing the comprehensiveness of the findings. Reproducibility was ensured by clearly reporting the methods and inclusion and exclusion criteria, allowing further researchers to replicate or update our review. The findings from various independent studies have provided a robust and evidence-based conclusion that helps strengthen current knowledge on the topic. Concerning the risk of bias, different tools, such as Rob-2 and NOS, were used to offer a clearer understanding of the reliability of evidence.

### 4.3. Limitations

Despite its strengths, this review also has several limitations that need to be considered. Considerable heterogeneity was observed among the included studies in terms of study populations and participants, which complicates the direct comparison of results. Studies with negative or even non-significant results are less likely to be published, increasing the potential publication bias. While RCTs report the specific type and dosage of each statin administered, this information was not reported in cohort studies. Additionally, definitional changes in sepsis criteria over time, particularly in studies conducted over extended periods, may have influenced outcomes and contributed to the variability or even inconclusiveness of earlier findings. Human error or subjective judgment cannot be entirely excluded, although efforts were made to minimize bias during study selection and data extraction.

### 4.4. Future Perspectives

While current evidence remains inconclusive, a deeper understanding of the underlying mechanisms may establish statins as supportive therapies in sepsis management. Originally developed for lipid lowering, statins also appear to exert immunomodulatory and anti-inflammatory effects that could be beneficial in sepsis. Ongoing and future clinical trials should aim to define the optimal dosing, timing, and patient selection to maximize these effects. Incorporating personalized medicine and biomarker-guided strategies may further enhance their therapeutic value. Together, these advances could expand the role of statins from cardiovascular prevention to critical care.

## 5. Conclusions

The present systematic review and meta-analysis demonstrates that statins, when combined with standard sepsis therapy, may enhance the therapeutic efficacy and contribute to improved clinical outcomes, by reducing mortality risk. Clinicians may consider the continuation or potential incorporation of statin use as an additional part in the treatment of septic patients.

## Figures and Tables

**Figure 1 medicina-61-01563-f001:**
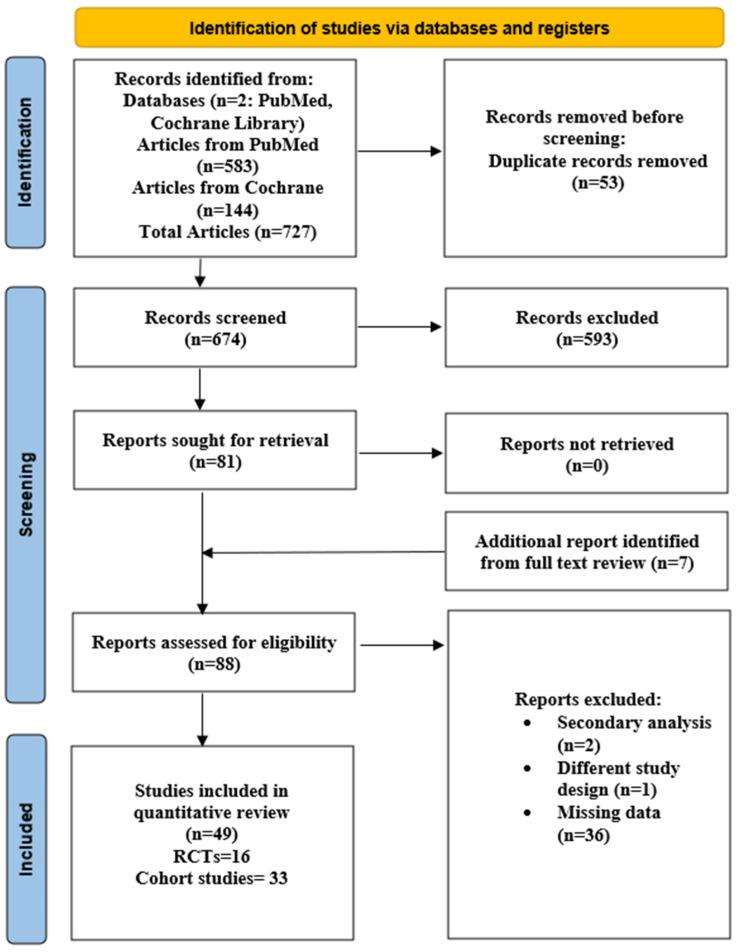
PRISMA flowchart of our systematic review.

**Figure 2 medicina-61-01563-f002:**
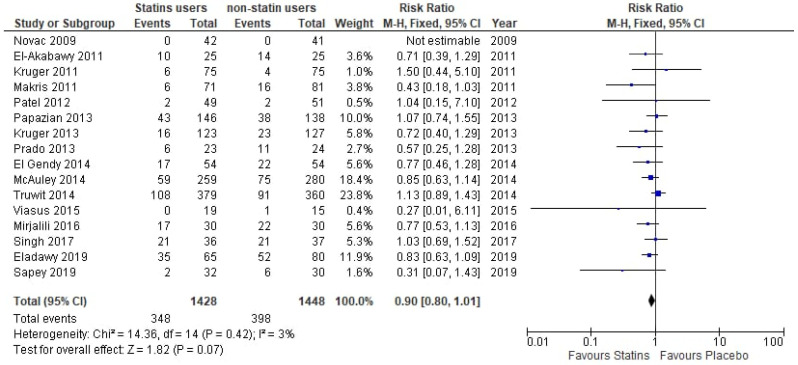
Forest plot of comparison: statin users vs. placebo in RCTs [16,17,18,19,20,21,22,23,24,25,26,27,28,29,30,31].

**Figure 3 medicina-61-01563-f003:**
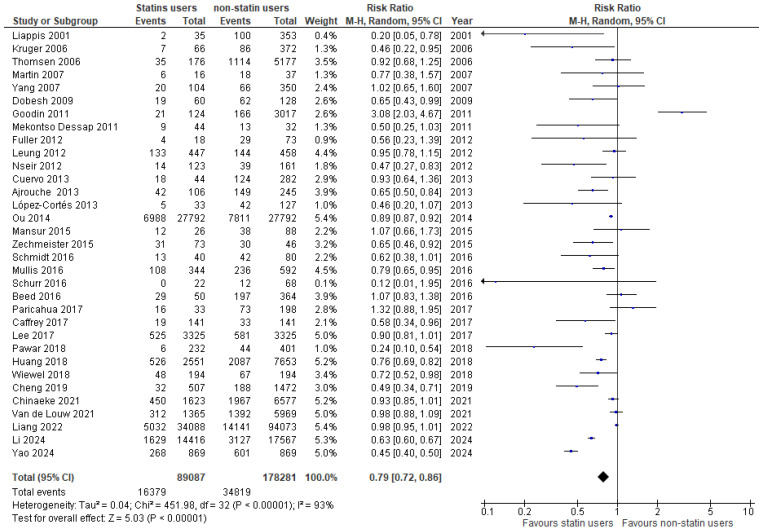
Forest plot of comparison: statin users vs. non-statin users in cohort studies [32,33,34,35,36,37,38,39,40,41,42,43,44,45,46,47,48,49,50,51,52,53,54,55,56,57,58,59,60,61,62,63,64].

**Figure 4 medicina-61-01563-f004:**
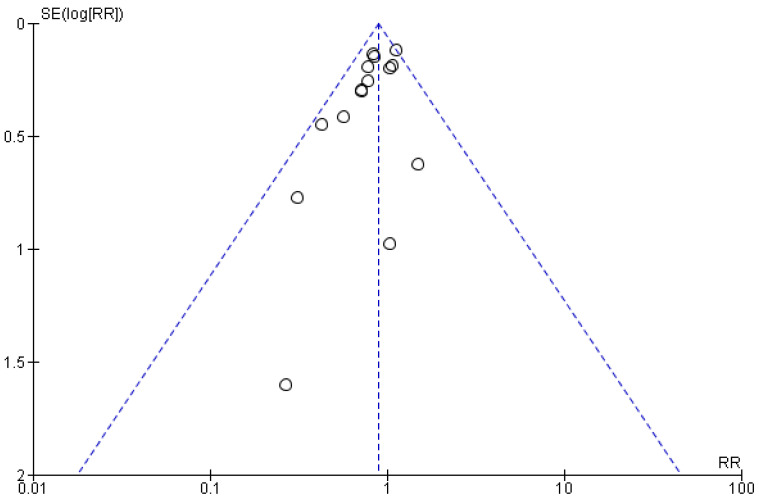
Funnel plot of comparison: statin users vs non-statin users in RCTs.

**Figure 5 medicina-61-01563-f005:**
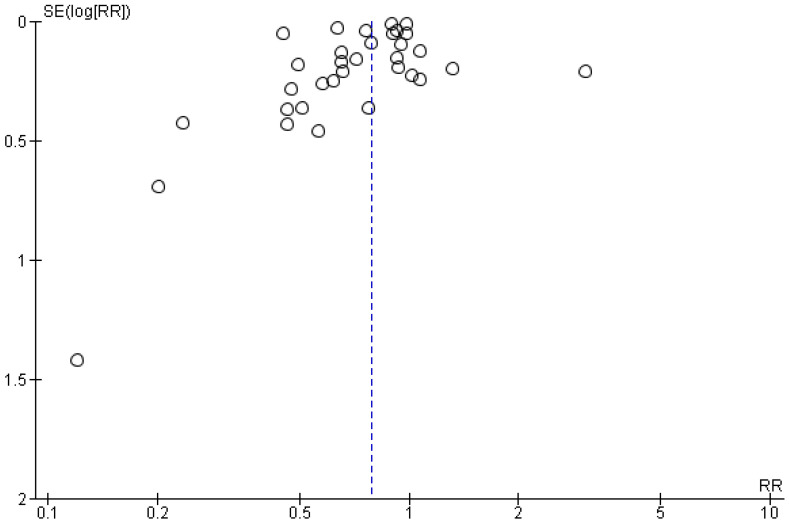
Funnel plot of comparison: statin users vs non-statin users in cohorts.

**Figure 6 medicina-61-01563-f006:**
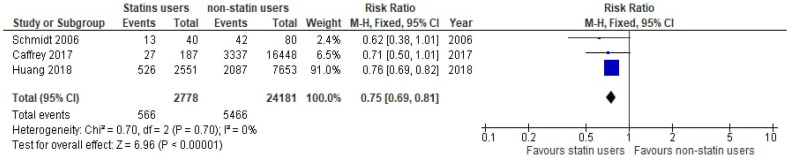
Forest plot of comparison: de novo statin use vs. non-statin use [34,55,58].

**Figure 7 medicina-61-01563-f007:**
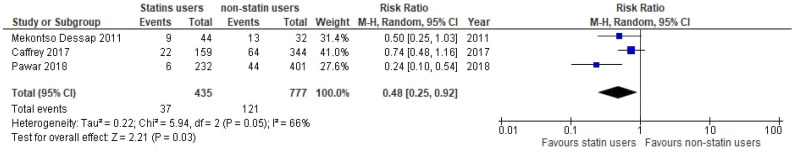
Forest plot of comparison: prior statin use and continuation vs. prior statin use and discontinuation [40,55,56].

**Figure 8 medicina-61-01563-f008:**
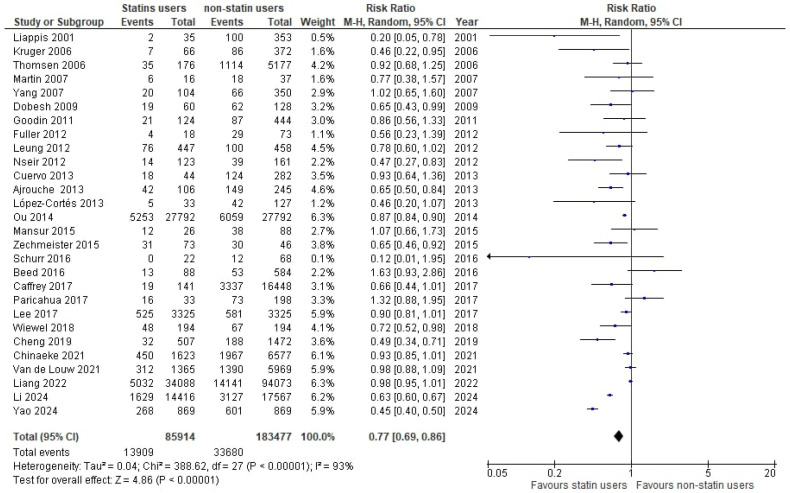
Forest plot of comparison: prior statin use and continuation of statins vs non-statin use [32,33,35,36,37,38,39,41,42,43,44,45,46,47,48,49,51,52,53,54,55,57,59,60,61,62,63,64].

**Figure 9 medicina-61-01563-f009:**
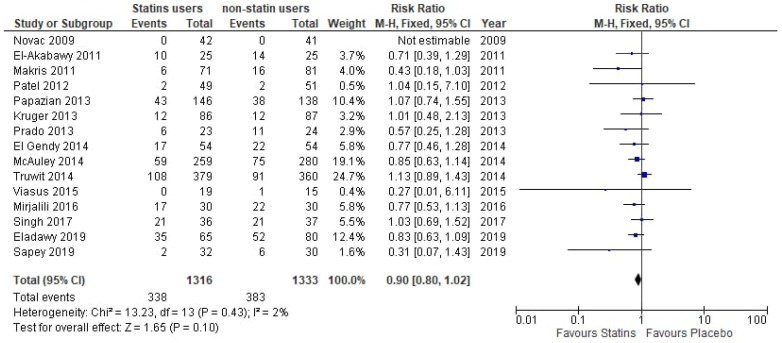
Forest plot of comparison: de novo statin use vs. placebo use [16,17,19,20,21,22,23,24,25,26,27,28,29,30,31].

**Figure 10 medicina-61-01563-f010:**
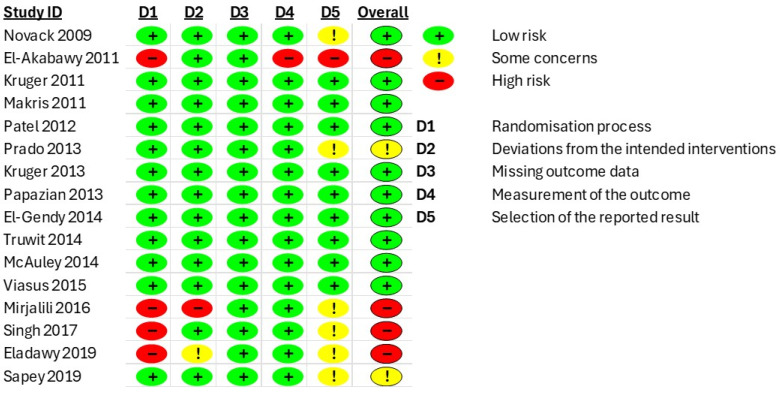
Risk of bias summary for RCTs [16,17,18,19,20,21,22,23,24,25,26,27,28,29,30,31].

**Figure 11 medicina-61-01563-f011:**
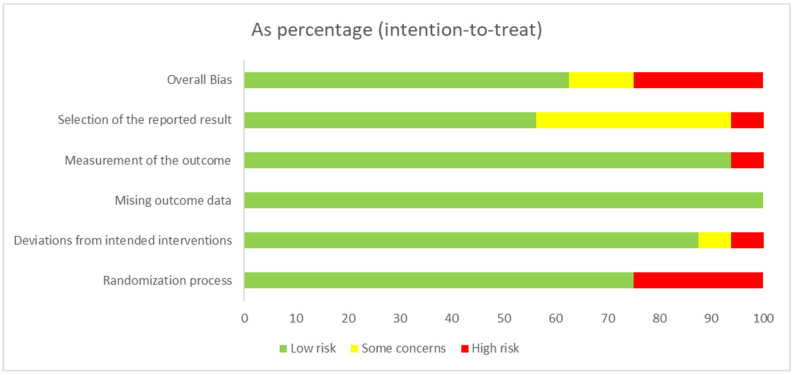
Summary plot displaying risk of bias assessment of included RCTs based on each domain of risk as described in the RoB2 tool.

**Figure 12 medicina-61-01563-f012:**
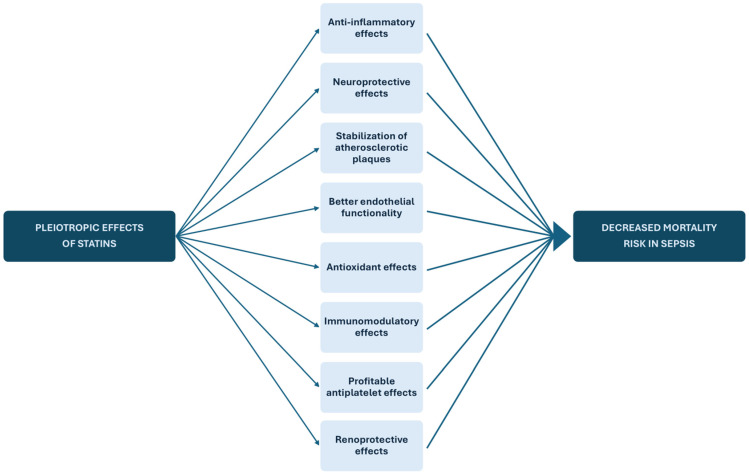
Pleiotropic effects of statins in sepsis.

**Table 1 medicina-61-01563-t001:** PICOS approach.

**P**opulation	Patients with sepsis
**I**ntervention	Statins
**C**omparator/controls	Placebo or no intervention
**O**utcomes	Mortality
**S**tudy design	Randomized control trials, case-control and cohort studies

**Table 2 medicina-61-01563-t002:** Study characteristics of included randomized control trials.

First Author	Year	Country	Study Design	Exposure and Comparator	Mortality (n, Statin Patients)	Mortality (n, Controls)	Statin	Dose of Statin	Underlying Condition
Novack [16]	2009	Israel	RCT	De novo statin use vs. placebo	0/42	0/41	Simvastatin	40 mg/day + 20 mg/day	Bacteremia
El-Akabawy [17]	2011	Egypt	RCT	De novo statin use vs. placebo	10/25	14/25	Atorvastatin	80 mg/day	Sepsis
Kruger [18]	2011	Australia	RCT	Prior statin use and continuation vs. prior statin use and placebo	6/75	4/75	Atorvastatin	20 mg/day	Bacteremia
Makris [19]	2011	Greece	RCT	De novo statin use vs. placebo	6/71	16/81	Pravastatin	40 mg/day	VAP
Patel [20]	2012	UK	RCT	De novo statin use vs. placebo	2/49	2/51	Atorvastatin	40 mg/day	Sepsis
Prado [21]	2013	Brazil	RCT	De novo statin use vs. placebo	6/23	11/24	Atorvastatin	80 mg/day	Sepsis/Septic shock
Kruger [22]	2013	Australia + New Zeeland	RCT	De novo statin use vs. placebo	12/86	12/87	Atorvastatin	20 mg/day	Sepsis
Papazian [23]	2013	France	RCT	De novo statin use vs. placebo	43/146	38/138	Simvastatin	60 mg/day	VAP
El Gendy [24]	2014	Egypt	RCT	De novo statin use vs. placebo	17/54	22/54	Rosuvastatin	20 mg/day	Sepsis
Truwit [25]	2014	USA	RCT	De novo statin use vs. placebo	108/379	91/360	Rosuvastatin	20 mg/day	SA-ARDS
McAuley [26]	2014	UK + Ireland	RCT	De novo statin use vs. placebo	59/259	75/280	Simvastatin	40 mg/day	ARDS
Viasus [27]	2015	Spain	RCT	De novo statin use vs. placebo	0/19	1/15	Simvastatin	20 mg/day	CAP
Mirjalili [28]	2016	Iran	RCT	De novo statin use vs. placebo	17/30	22/30	Simvastatin	40 mg/day	Sepsis
Singh [29]	2017	India	RCT	De novo statin use vs. placebo	21/36	21/37	Atorvastatin	40 mg/day	Septic shock
Eladawy [30]	2019	Egypt	RCT	De novo statin use vs. placebo	35/65	52/80	Rosuvastatin	20 mg/day	Sepsis
Sapey [31]	2019	UK	RCT	De novo statin use vs. placebo	2/32	6/30	Simvastatin	40 mg/day	Pneumonia

ARDS: acute respiratory distress syndrome; CAP: community-associated pneumonia; SA-ARDS: sepsis-associated ARDS; RCT: randomized controlled trial; VAP: ventilator-associated pneumonia.

**Table 3 medicina-61-01563-t003:** Study characteristics of included cohort studies.

First Author	Year	Country	Study Design	Exposure and Comparator	Mortality (n/Statin Patients)	Mortality (n/Controls)	Underlying Condition
Liappis [32]	2001	USA	Retrospective Cohort study	Prior statin use + continuation vs. non-statin users	2/35	100/353	Bacteremia
Thompsen [33]	2006	Denmark	Retrospective Cohort study	Prior statin use + continuation vs. non-statin users	35/176	1114/5177	Bacteremia
Schmidt [34]	2006	Germany	Retrospective Cohort study	De novo statin use vs. non-statin users	13/40	42/80	MODS
Kruger [35]	2006	UK	Retrospective Cohort study	Prior statin use + continuation vs. non-statin users	7/66	86/372	Bacteremia
Martin [36]	2007	USA	Retrospective Cohort study	Prior statin use + continuation vs. non-statin users	6/16	18/37	Sepsis
Yang [37]	2007	Taiwan	Retrospective Cohort study	Prior statin use + continuation vs. non-statin users	20/104	66/350	Sepsis
Dobesh [38]	2009	USA	Retrospective Cohort study	Prior statin use + continuation vs. non-statin users	19/60	62/128	Severe sepsis
Goodin [39]	2011	USA	Retrospective Cohort study	Prior statin use + continuation vs. non-statin users	21/124	87/444	Sepsis
Mekontso Dessap [40]	2011	France	Retrospective Cohort study	Prior statin + continuation vs. prior statin + non-continuation	9/44	13/32	Severe Sepsis + Septic shock
Leung [41]	2012	USA	Retrospective Cohort study	Prior statin use + continuation vs. non-statin users	76/447	100/458	Bacteremia
Fuller [42]	2012	USA	Retrospective Cohort study	Prior statin use + continuation vs. non-statin users	4/18	29/73	Septic shock
Nseir [43]	2012	Israel	Retrospective Cohort study	Prior statin use + continuation vs. non-statin users	14/123	39/161	Bacteremia
Cuervo [44]	2013	Argentina + Brazil + Spain	Retrospective Cohort study	Prior statin use + continuation vs. placebo	18/44	124/282	Candidemia
Ajrouche [45]	2013	Lebanon	Retrospective Cohort study	Prior statin use + continuation vs. non-statin users	42/106	149/245	Sepsis
Lopez-Cortes [46]	2013	Spain	Retrospective Cohort study	Prior statin use + continuation vs. non-statin users	5/33	42/127	Bacteremia
Ou [47]	2014	Taiwan	Retrospective Cohort study	Prior statin use + continuation vs. non-statin users	5253/27,792	6059/27,792	Sepsis
Zechmeister [48]	2015	USA	Retrospective Cohort study	Prior statin use + continuation vs. non-statin users	31/73	30/46	Septic shock
Mansur [49]	2015	Germany	Retrospective Cohort study	Prior statin use + continuation vs. non-statin users	12/26	38/88	ARDS
Mullis [50]	2016	USA	Retrospective Cohort study	Statin vs. placebo	108/344	236/592	Sepsis
Beed [51]	2016	UK	Retrospective Cohort study	Prior statin use + continuation vs. non-statin users	13/88	53/584	Sepsis
Schurr [52]	2016	USA	Retrospective Cohort study	Prior statin use + continuation vs. non-statin users	0/22	12/68	Sepsis
Paricahua [53]	2017	Argentina	Retrospective Cohort study	Prior statin use + continuation vs. non-statin users	16/33	73/198	Sepsis
Lee [54]	2017	Taiwan	Retrospective Cohort study	Prior statin use + continuation vs. non-statin users	525/3325	581/3325	Severe Sepsis
Caffrey [55]	2017	USA	Retrospective Cohort study	Prior statin use + continuation vs. non-statin users	19/141	33/141	Bacteremia
Pawar [56]	2018	USA	Retrospective Cohort study	Prior statin + continuation vs. prior statin + non-continuation	6/232	44/401	Bacteremia
Wiewel [57]	2018	Netherlands	Retrospective Cohort study	Prior statin use + continuation vs. non-statin users	48/194	67/194	Sepsis
Huang [58]	2018	Taiwan	Retrospective Cohort study	De novo statin use vs. non-statin users	526/2551	2087/7653	CKD
Cheng [59]	2019	Taiwan	Retrospective Cohort study	Prior statin use + continuation vs. non-statin users	32/507	188/1472	Bacteremia
Chinaeke [60]	2021	USA	Retrospective Cohort study	Prior statin use + continuation vs. non-statin users	450/1623	1967/6577	Sepsis
Van de Louw [61]	2021	USA	Retrospective Cohort study	Prior statin use + continuation vs. non-statin users	312/1365	1390/5969	Sepsis
Liang [62]	2022	USA	Retrospective Cohort study	Prior statin use + continuation vs. non-statin users	5032/34,088	14,141/94,073	Sepsis
Yao [63]	2024	China	Retrospective Cohort study	Prior statin use + continuation vs. non-statin users	268/869	601/869	Coagulopathy
Li [64]	2024	Netherlands	Retrospective Cohort study	Prior statin use + continuation vs. non-statin users	1629/14,416	3127/17,567	Sepsis

ARDS: acute respiratory distress syndrome; CKD: chronic kidney disease; MODS: multi-organ dysfunction syndrome.

**Table 4 medicina-61-01563-t004:** Quality appraisal of the included studies using the Newcastle–Ottawa Scale (NOS) for cohort studies.

Study	Selection				Comparability	Outcome			Quality
	Representativeness of the Cohort	Non-Exposed Cohort Selection	Exposure Ascertainment	Outcome of Interest		Assessment	Follow-Up Time	Follow-Up Adequacy	
Liappis et al.， 2001 [32]	★	★	★	★	★★	★	-	-	7
Thompsen et al.， 2006 [33]	★	★	★	★	★★	★	★	-	8
Schmidt et al.， 2006 [34]	-	★	★	★	★★	★	★	-	7
Kruger et al.， 2006 [35]	★	★	★	★	★★	★	-	-	7
Martin et al.， 2007 [36]	★	★	★	★	★★	★	-	-	7
Yang et al.， 2007 [37]	★	★	★	★	★★	★	-	-	7
Dobesh et al.， 2009 [38]	★	★	★	★	★★	★	-	-	7
Goodin et al.， 2011 [39]	★	★	★	★	★★	★	-	-	7
Mekontso Dessap et al.， 2011 [40]	★	★	★	★	★★	★	-	-	7
Leung et al.， 2012 [41]	★	★	★	★	★★	★	-	-	7
Fuller et al.， 2012 [42]	★	★	★	★	★★	★	-	-	7
Nseir et al.， 2012 [43]	★	★	★	★	★★	★	-	-	7
Ajrouche et al.， 2013 [45]	★	★	★	★	★★	★	-	-	7
Lopez-Cortes et al.， 2013 [46]	★	★	★	★	★★	★	★	-	8
Cuervo et al.， 2013 [44]	★	★	★	★	★★	★	-	-	7
Ou et al.， 2014 [47]	★	★	★	★	★★	★	★	★	9
Zechmeister et al.， 2015 [48]	★	★	★	★	★★	★	-	-	7
Mansur et al.， 2015 [49]	★	★	★	★	★★	★	★	★	9
Mullis et al.， 2016 [50]	★	-	★	★	-	★	★	★	6
Beed et al.， 2016 [51]	★	★	★	★	★★	★	★	★	9
Schurr et al.， 2016 [52]	★	★	★	★	★★	★	-	-	7
Paricahua et al.， 2017 [53]	★	★	★	★	★★	★	-	-	7
Lee et al.， 2017 [54]	★	★	★	★	★★	★	-	-	7
Caffrey et al.， 2017 [55]	-	-	-	★	★★	★	-	-	4
Pawar et al.， 2018 [56]	★	★	★	★	★★	★	-	-	7
Wiewel et al.， 2018 [57]	★	★	★	★	★★	★	-	-	7
Huang et al.， 2018 [58]	★	★	★	★	★★	★	★	-	8
Cheng et al.， 2019 [59]	★	★	★	★	★★	★	-	-	7
Chinaeke et al.， 2021 [60]	★	★	★	★	★★	★	-	-	7
Van de Louw et al.， 2021 [61]	★	★	★	★	★★	★	★	★	9
Liang et al.， 2022 [62]	★	★	★	★	★★	★	★	★	9
Yao et al.， 2024 [63]	★	★	★	★	★★	★	-	-	7
Li et al.， 2024 [64]	★	★	★	★	★★	★	-	-	7

The maximum score that can be given to each included study is 9 points; A score of 7–9 stars indicates good, a score of 4–6 stars indicates fair, while a score of 0–3 shows poor quality studies.

## Data Availability

The original contributions presented in this study are included in the article. Further inquiries can be directed to the corresponding author.

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
