# Peer review of "The Impact of Statin Use on Sepsis Mortality: A Systematic Review and Meta-Analysis"

_medicina, 2025, doi:10.3390/medicina61091563_

Round 1
Reviewer 1 Report
Comments and Suggestions for Authors
In this systematic review and meta-analysis, the authors investigated the potential of statins in reducing sepsis mortality. Overall, this is an interesting study with meaningful clinical implications. However, I have several suggestions and points for clarification:
- Consider adding a diagram illustrating the pleiotropic effects of statins in sepsis (as discussed in Section 4.1).
- Statins are primarily prescribed for ASCVD and metabolic conditions. Including patients with these diseases may confound the results. The authors should either exclude such patients or perform subgroup/sensitivity analyses to account for this.
- In sepsis, early and late mortality have different mechanisms and risk factors. Please clarify throughout the manuscript (including the title, abstract, and text) which phase of mortality was analyzed.
- In Section 2.6, please specify whether a random-effects or fixed-effects model was applied.
- It would be helpful to distinguish retrospective from prospective cohorts. For instance, the PRISMA flowchart and tables should indicate the number of studies in each category.
- Please revise the format of confidence intervals as follows: 95% CI 0.72–0.86.
- The statement "statin users have 10% lower mortality risk … with a trend toward significance (p = 0.07)" is misleading. Since p > 0.05, it should be reported as non-significant without implying a trend toward or away from significance.
- In the pooled analysis of cohort studies (Figure 3), the heterogeneity (I² = 93%) is substantial. A sensitivity analysis excluding potential outlier studies is recommended to assess the robustness of the findings.
- Section 3.4 should be titled “Subgroup Analyses,” not “Sensitivity Analyses,” as the content pertains to subgroup comparisons.
- Please perform sensitivity analyses for Figure 8 to determine if the pooled result remains consistent.
Author Response
Reviewer 1
In this systematic review and meta-analysis, the authors investigated the potential of statins in reducing sepsis mortality. Overall, this is an interesting study with meaningful clinical implications. However, I have several suggestions and points for clarification:
Response: The authors thank the reviewer for their positive remarks. We hope we have adequately responded to reviewer’s suggestions and comments.
- Consider adding a diagram illustrating the pleiotropic effects of statins in sepsis (as discussed in Section 4.1).
Response: As suggested, please see below the illustration, which is added in the manuscript as Figure 14.
Figure 14. Pleiotropic effects of statins in sepsis.
- Statins are primarily prescribed for ASCVD and metabolic conditions. Including patients with these diseases may confound the results. The authors should either exclude such patients or perform subgroup/sensitivity analyses to account for this.
Response: The authors thank the reviewer for this comment. Indeed, statins are the cornerstone of hypolipidemic treatment for primary and secondary ASCVD prevention. There were not enough data to perform further analysis regarding confounding factors. However, we did perform a meta-analysis on patients already on statins treatment. The result of this sensitivity analysis on patients already on statin use and continuation vs prior statin use and discontinuation are shown in Figure 7.
- In sepsis, early and late mortality have different mechanisms and risk factors. Please clarify throughout the manuscript (including the title, abstract, and text) which phase of mortality was analyzed.
Response: We thank the reviewer for this comment. In our analysis we included the overall mortality either this was 14 days, 28 days, 30 days, 60 days, or 90 days. The majority of the 16 included RCTs, however, reported 28-30 day mortality risk (10 studies), 1 study reported 14 day mortality, 1 study 60 day mortality, 1 study 90 day mortality, while 3 studies did not mention the phase of mortality. In the 33 included cohorts, 13 studies reported 28-30 day mortality, while the rest 20 studies did not mention any phase regarding mortality.
We have added the above information in the “Study and population characteristics” section (section 3.2), as follows:
Of note, the majority of the included RCTs, reported 28-day or 30-day mortality (10 studies), 1 study reported 14-day mortality, 1 study reported 60-day mortality, 1 study reported 90-day mortality, while 3 studies did not mention how mortality was evalu-ated. In the 33 included cohorts, 13 studies reported 28-day or 30-day day mortality, while 20 studies did not mention details regarding mortality.
We did not mention the above information on the title and abstract, as this information was missing information in several studies.
- In Section 2.6, please specify whether a random-effects or fixed-effects model was applied.
Response: We added the following sentence in paragraph 2.6:
“The random-effects model was used when heterogeneity was high and fixed-effects model was used in the performed meta-analyses when the heterogeneity was low.
- It would be helpful to distinguish retrospective from prospective cohorts. For instance, the PRISMA flowchart and tables should indicate the number of studies in each category.
Response: All cohorts are retrospective. This is now mentioned in table 3.
- Please revise the format of confidence intervals as follows: 95% CI 72–0.86.
Response: Done.
- The statement "statin users have 10% lower mortality risk … with a trend toward significance (p = 0.07)" is misleading. Since p > 0.05, it should be reported as non-significant without implying a trend toward or away from significance.
Response: The authors thank the reviewer for this comment. The “trend towards significance” was removed from the abstract, and the results sections, and changed as follows in the discussion part:
“…while the pooled analysis of RCTs demonstrated a non-significant alteration of mortality risk.”
- In the pooled analysis of cohort studies (Figure 3), the heterogeneity (I² = 93%) is substantial. A sensitivity analysis excluding potential outlier studies is recommended to assess the robustness of the findings.
Response: After excluding the outlier studies by “Liappis, Pawar, Goodin, Ou, Schurr, Huang, Liang, and Li” the heterogeneity remained considerably high (87%). Please see below the pooled analysis that was added:
We have added the following paragraph to the results section:
3.5. Sensitivity Analyses
Due to high heterogeneity, we also performed sensitivity analyses for figures 3 and 8. Regarding Figure 3, after excluding the outlier studies by Liappis, Pawar, Goodin, Ou, Schurr, Huang, Liang, and Li the heterogeneity remained considerably high (87%), (Figure 10). Similarly, regarding Figure 8, after excluding the outlier studies by Liappis, Fuller, Goodin, Ou, Schurr, Beed, Li, Liang and Yao, the heterogeneity remained relatively high (61%), (Figure 11).
- Section 3.4 should be titled “Subgroup Analyses,” not “Sensitivity Analyses,” as the content pertains to subgroup comparisons.
Response: Done
- Please perform sensitivity analyses for Figure 8 to determine if the pooled result remains consistent.
Response: After excluding the outlier studies by “Liappis, Fuller, Goodin, Ou, Schurr, Beed, Li, Liang and Yao” the heterogeneity remained relatively high (61%). Please see below the pooled analysis that was added through the text:
We have added the following paragraph to the results section:
3.5. Sensitivity Analyses
Due to high heterogeneity, we also performed sensitivity analyses for figures 3 and 8. Regarding Figure 3, after excluding the outlier studies by Liappis, Pawar, Goodin, Ou, Schurr, Huang, Liang, and Li, heterogeneity remained considerably high (87%), (Figure 10). Similarly, regarding Figure 8, after excluding the outlier studies by Liappis, Fuller, Goodin, Ou, Schurr, Beed, Li, Liang and Yao, heterogeneity remained relatively high (61%) (Figure 11).
Please see the attachment

Reviewer 2 Report
Comments and Suggestions for Authors
The manuscript " The impact of statin use on sepsis mortality: A Systematic Review and Meta-analysis" in which the authors evaluated the current evidence regarding the potential protective effects of statins on sepsis mortality." They found that statin use among septic patients is associated with a reduction in mortality, suggesting that statins may offer a beneficial therapeutic effect in the clinical setting.
The work is understandable and important. However, this paper suffers from some shortcomings that should be modified.
Shortcomings
1- In Abstract section, you don't need to add the level of significance in the abstract part.
2- In Introduction section, please add the standard drugs used in treatment of sepsis and list some advantages and disadvantages of these drugs suggesting the need to search for other drugs such as statin.
3- In Materials and methods, please add the measure of blinding in this study.
4- In Discussion section,
*Please discuss in detail, the molecular mechanisms that could be targeted by statin in patients with sepsis.
*Please discuss briefly , the pharmacokinetics, safety, possible adverse effects of statins in human.
*Please discuss briefly, the suitability and drug interaction of combining statins with the standard drugs of sepsis.
5- Please add your future perspectives.
Author Response
Reviewer 2
The manuscript " The impact of statin use on sepsis mortality: A Systematic Review and Meta-analysis" in which the authors evaluated the current evidence regarding the potential protective effects of statins on sepsis mortality." They found that statin use among septic patients is associated with a reduction in mortality, suggesting that statins may offer a beneficial therapeutic effect in the clinical setting.
The work is understandable and important. However, this paper suffers from some shortcomings that should be modified.
Response: The authors thank the reviewer for commenting positively on our manuscript. We hope that our responses below are of sufficient quality.
Shortcomings
1- In Abstract section, you don't need to add the level of significance in the abstract part.
Response: We thank the reviewer for this comment. We have removed all p values from the abstract.
2- In Introduction section, please add the standard drugs used in treatment of sepsis and list some advantages and disadvantages of these drugs suggesting the need to search for other drugs such as statin.
Response: We appreciate the reviewer’s comment. We added the following paragraph to the introduction section:
Early fluid resuscitation, broad spectrum antibiotics, vasopressors, oxygen sup-ple-mentation and corticosteroids in certain cases are among the standard regimens of sepsis [2]. While these interventions are essential and beneficial, sepsis mortality rates still remain high [3]. As a result, it is crucial to explore alternative agents, such as statins, and examine whether those agents exert beneficial effects on sepsis or not.
3- In Materials and methods, please add the measure of blinding in this study.
Response: We have added this information in the “2.5. Assessment of bias” subsection.
4- In Discussion section,
*Please discuss in detail, the molecular mechanisms that could be targeted by statin in patients with sepsis.
*Please discuss briefly , the pharmacokinetics, safety, possible adverse effects of statins in human.
*Please discuss briefly, the suitability and drug interaction of combining statins with the standard drugs of sepsis.
Response: The authors appreciate the reviewer’s comment. We have added the following paragraphs in discussion subsection 4.1. highlighted in yellow.
4.1. Pathophysiological Mechanisms of Statins Action on Sepsis
In an attempt to clarify the role of statins in sepsis mortality, understanding the underlying pathophysiological mechanisms by which statins influence the progression of disease is equally important. Concerning the pleiotropic effects of statins (Figure 14), several studies have identified a reduction of the inflammatory process, neuroprotective properties and improvement in kidney function. Inhibition of the enzyme HMG-CoA reductase reduces the mevalonate pathway intermediates like farnesyl, which impairs the prenylation of GTPases and lowers the pro-inflammatory signaling [67]. Additionally, statins increase the stabilization of atherosclerotic plaques, exert beneficial actions on endothelial functionality, offer antioxidant, anti-inflammatory and immunomodulatory effects and provide profitable antiplatelet properties [4]. It has been found that the improvement in endothelial vasoreactivity is mainly linked to reduced oxidative stress and drug-induced vasorelaxation. Statins help restore the balance between nitric oxide (NO) and reactive oxygen species (ROS). They enhance endothelial NO synthase (eNOS) activity and NO bioavailability, thus improving vasodilation and endothelial function while reducing oxidative degradation of NO [68]. Moreover, experimental animal studies have determined that statin therapy not only attenuates the production of superoxide anion (O−2) and NADPH oxidase but also decreases the endogenous peroxides and peroxidase activity. In combination with the production of haem oxygenase, an enzyme with antioxidant properties, it has been established that statins may exert antioxidant properties [68,69]. Other than that, statin use has been shown to minimize the levels of inflammatory markers. These include, among others, high-sensitivity C-reactive protein (hs-CRP), fibrinogens, serum amyloid A, von Willebrand factor (vWF) and platelet-activating factor acetylhydrolase (PAFAH). This is accompanied by the attenuation of several proinflammatory cytokines, such as tumor necrosis factor-α (TNF-α), interferon-γ (IFN-γ) and Interleukin 6 and 8 (IL-6, IL-8), by suppressing NF-κB and TLR4 activation, providing clinical evidence of anti-inflammatory and immunomodulatory effects of statins [8,70]. Statin treatment may also exert antiplatelet properties by inhibiting the aggregation of leukocytes. This is achieved by decreasing tissue factor expression and increasing thrombomodulin activity, helping prevent microvascular thrombosis and ischemia [71]. Additionally, its neuroprotective effects mediated through increased eNOS expression and Akt activation and phosphorylation, along with its renoprotective effects, such as the reduction of proteinuria, provide a broader non-cardiovascular benefit of statin therapy [4,72,73].
Statins are orally administered and undergo substantial first-pass hepatic uptake via OATP1B1, with clearance mainly through biliary excretion with variable renal contribution depending on the agent [74]. Metabolism is agent-specific; simvastatin, lovastatin and atorvastatin are CYP3A4 substrates, fluvastatin is mainly CYP2C9, while pitavastatin, rosuvastatin and pravastatin have minimal CYP metabolism. Half-life of statins is again class-specific, making evening dosing for short-acting and flexible timing for longer-acting ones preferable. All statins are generally very well tolerated [4]. Myalgia, headache and gastrointestinal discomfort are the most common complaints, while myopathy, rhabdomyolysis and transaminase elevation represent significant adverse effects. Regarding drug-drug interactions, CYP3A3 inhibitors (e.g. macrolides, protease inhibitors) and grapefruit juice increase exposure and toxicity risk [74].
Overall, there are no major concerns regarding potential interactions between statins and standard sepsis treatment. The only exceptions concern macrolides and azoles, which can raise statin levels causing toxicity. In patients with acute liver injury or rhabdomyolysis risk, statin use should be deferred until renal and hepatic function is within normal function [75].
5- Please add your future perspectives.
Response: We have added the following subsection in the Discussion:
4.4. Future perspectives
While current evidence remains inconclusive, a deeper understanding of the underlying mechanisms may establish statins as supportive therapies in sepsis management. Originally developed for lipid lowering, statins also appear to exert immunomodulatory and anti-inflammatory effects that could be beneficial in sepsis. Ongoing and future clinical trials should aim to define the optimal dosing, timing, and patient selection to maximize these effects. Incorporating personalized medicine and biomarker-guided strategies may further enhance their therapeutic value. Together, these advances could expand the role of statins from cardiovascular prevention to critical care.
Please see the attachment

Reviewer 3 Report
Comments and Suggestions for Authors
This manuscript presents a well-structured and comprehensive systematic review and meta-analysis of the impact of statin therapy on sepsis mortality, synthesizing data from both RCTs and cohort studies. The topic is highly relevant as sepsis remains a leading cause of hospital mortality, and statins are commonly prescribed medications. The inclusion of 49 studies with over 270,000 participants provides substantial power to the conclusions. Differentiating between de novo use, prior continuation, and comparisons with placebo enhances interpretability.
My comments are:
- Although mentioned as a limitation, the heterogeneity seems high. Consider adding subgroup or meta-regression analyses to identify potential sources of heterogeneity.
- Unlike RCTs, the cohort studies did not report the type or dosage of statins used. This limits conclusions regarding whether specific statins or intensities may be more effective. This needs to be acknowledged in the limitations section.
- It is possible that some of these studies to have taken a long time, with the possibility of the sepsis criteria shifting. Discuss how definitional changes over time might have influenced the findings and how they were handled during study selection.
Author Response
Reviewer 3
This manuscript presents a well-structured and comprehensive systematic review and meta-analysis of the impact of statin therapy on sepsis mortality, synthesizing data from both RCTs and cohort studies. The topic is highly relevant as sepsis remains a leading cause of hospital mortality, and statins are commonly prescribed medications. The inclusion of 49 studies with over 270,000 participants provides substantial power to the conclusions. Differentiating between de novo use, prior continuation, and comparisons with placebo enhances interpretability.
Response: The authors thank the reviewer for stating that this is a well-structured and comprehensive systematic review and meta-analysis. Please see our point-by-point responses below, which we hope are sufficient.
My comments are:
- Although mentioned as a limitation, the heterogeneity seems high. Consider adding subgroup or meta-regression analyses to identify potential sources of heterogeneity.
Response: We have added the following subsection to the results:
3.5. Sensitivity Analyses
Due to high heterogeneity, we also performed sensitivity analyses for figures 3 and 8. Regarding Figure 3, after excluding the outlier studies by Liappis, Pawar, Goodin, Ou, Schurr, Huang, Liang, and Li the heterogeneity remained considerably high (87%), (Figure 10). Similarly, regarding Figure 8, after excluding the outlier studies by Liappis, Fuller, Goodin, Ou, Schurr, Beed, Li, Liang and Yao, the heterogeneity remained relatively high (61%), (Figure 11).
Figure 10. Forest plot of comparison: Statin users vs non-Statin users in Cohort studies, after excluding the outlier studies.
Figure 11. Forest plot of comparison: prior statin use and continuation of statins vs non-statin use, after excluding the outlier studies.
- Unlike RCTs, the cohort studies did not report the type or dosage of statins used. This limits conclusions regarding whether specific statins or intensities may be more effective. This needs to be acknowledged in the limitations section.
Response: We thank the reviewer for the comment. We added the following sentence to “Limitations” section:
“While RCTs report the specific type and dosage of each statin administered, this information was not reported in cohort studies.”
- It is possible that some of these studies to have taken a long time, with the possibility of the sepsis criteria shifting. Discuss how definitional changes over time might have influenced the findings and how they were handled during study selection.
Response: We thank the reviewer for the comment. We added the following sentence to “Limitations” section:
“Additionally, definitional changes in sepsis criteria over time, particularly in studies conducted over extended periods, may have influenced outcomes and contributed to the variability or even inconclusiveness of earlier findings.”
Please see the attachment

Round 2
Reviewer 1 Report
Comments and Suggestions for Authors
Thank you for the response, I would like to comment further regarding the sensitivity analyses:
Sensitivity analysis (one leave out study; https://jbi-global-wiki.refined.site/space/MANUAL/355828467/4.3.8+Sensitivity+analysis+in+meta-analysis) was meant to find which studies causes high heterogeneity and therefore should be excluded to see whether the pooled value remain. Therefore, please re-perform this analysis and attach all the forrest plot as supplementary figures. In section 3.5, the authors must find which study(ies) contribute to this and check the pooled value. At this point everything is unclear and section 3.5 is meaningless.
